# The Role of Necroptosis: Biological Relevance and Its Involvement in Cancer

**DOI:** 10.3390/cancers13040684

**Published:** 2021-02-08

**Authors:** Laura Della Torre, Angela Nebbioso, Hendrik G. Stunnenberg, Joost H. A. Martens, Vincenzo Carafa, Lucia Altucci

**Affiliations:** 1Department of Precision Medicine, Università Degli Studi Della Campania “Luigi Vanvitelli”, 80138 Naples, Italy; laura.dellatorre@unicampania.it (L.D.T.); angela.nebbioso@unicampania.it (A.N.); 2Department of Molecular Biology, Faculty of Science, Radboud Institute for Molecular Life Sciences, Radboud University, 6525 GA Nijmegen, The Netherlands; 3Princess Maxima Center for Pediatric Oncology, 3584 CS Utrecht, The Netherlands; H.G.Stunnenberg@prinsesmaximacentrum.nl

**Keywords:** necroptosis, cancer, RIPK1, RIPK1 mutations

## Abstract

**Simple Summary:**

A new form of programmed necrosis called necroptosis has emerged. This new and well-documented type of programmed cell death is involved in several human diseases, including cancer. RIPK1, the main mediator of necroptosis, in response to different stimuli, activates several molecular pathways leading to inflammation, cell survival, or cell death. Targeting necroptosis could be a new strategy for advanced therapies. In this review, we focus on the biological relevance of this type of programmed cell death and its main executor RIPK1 in pathogenesis to find novel potential clinical intervention strategies.

**Abstract:**

Regulated cell death mechanisms are essential for the maintenance of cellular homeostasis. Evasion of cell death is one of the most important hallmarks of cancer. Necroptosis is a caspase independent form of regulated cell death, investigated as a novel therapeutic strategy to eradicate apoptosis resistant cancer cells. The process can be triggered by a variety of stimuli and is controlled by the activation of RIP kinases family as well as MLKL. The well-studied executor, RIPK1, is able to modulate key cellular events through the interaction with several proteins, acting as strategic crossroads of several molecular pathways. Little evidence is reported about its involvement in tumorigenesis. In this review, we summarize current studies on the biological relevance of necroptosis, its contradictory role in cancer and its function in cell fate control. Targeting necroptosis might be a novel therapeutic intervention strategy in anticancer therapies as a pharmacologically controllable event.

## 1. Introduction

Proliferation, differentiation, and cell death are physiological events responsible for the maintenance of cellular homeostasis [1]. Each cell has a well-defined life cycle at the end of which it dies naturally [2]. Cell death can occur in two different modalities, one following a passive, uncontrolled or accidental process, the other one through an active and regulated process [3].

Cells die not only as a physiological phenomenon but also when exposed to chemical, physical, and mechanical insults and, based on the severity of the trauma, we distinguish accidental cell death (ACD), a catastrophic and extreme event with consequent disruption of cellular structure since it is completely unprogrammed, with regulated cell death (RCD), which is a controlled process, depending on activation of signaling cascades and hence can be genetically and pharmacologically controlled [4,5,6].

When RCD occurs in a physiological scenario with the preservation of development and tissue homeostasis, it is known as programmed cell death (PCD) [3,5] (Figure 1).

Apoptosis and necrosis were the first and so the best known two modalities of cell death discovered. Historically, apoptosis has been associated with non-immunogenic, regulated, and programmed forms of death, while necrosis has been defined as accidental and uncontrolled [6]. Apoptosis is a defense mechanism that occurs physiologically as a result of extracellular or intracellular microenvironment perturbations and is mediated by a specific class of cysteine protease named caspases [7]. After extracellular or intracellular apoptotic stimuli, caspases activate all molecular pathways necessary to eliminate damaged cells [8]. Conversely, necrosis is the result of accidental events induced by different stimuli such as osmotic variations, arrest of the supply of nutrients, protein denaturation and it is characterized by a robust inflammatory and immunogenic response [5,9].

Referring only to morphological criteria, cell death was initially classified as (i) apoptosis, PCD type I; (ii) autophagy, PCD type II; and (iii) necrosis, lacking PCD type I and II features [2]. Following advancements in biochemical approaches, the Nomenclature Committee on Cell Death (NCCD) has updated RCDs guidelines, by the addition and description of other forms of programmed cell death, whose molecular mechanisms and biological functions must be better clarified [5,6,10]. This classification subdivided the RCD into 12 classes on the basis of molecular mechanisms, morphological features, and immunomodulatory profile: intrinsic apoptosis, extrinsic apoptosis, MPT-driven necrosis, necroptosis, ferroptosis, pyroptosis, parthanatos, entotic cell death, NETotic cell death, lysosome-dependent cell death, autophagy-dependent cell death, and immunogenic cell death [11]. It is now known that there is a molecular interconnectivity between the different RCDs, although each one is characterized by a specific signal transduction cascade and morphological, biochemical, and immunogenic features as biomarkers [11].

Among them, necroptosis is the first form of programmed necrosis described with a prominent role in multiple physiological and pathological conditions [6]. It is induced by different extracellular or intracellular stimuli and detected by specific death receptors, PRRs and ZBP1 [10,11,12]. Numerous evidences suggest the involvement of necroptosis in the pathogenesis of diseases including organ injury, neurodegenerative diseases and viral infections [12,13]. Furthermore, necroptosis seems to control the homeostasis of T cells by eliminating the abnormal and excess ones in caspase-8 deficient T cells [14]. In addition to these physio-pathological functions, necroptosis shows a contradictory role in cancer that needs to be better investigated. Indeed, several studies suggest its involvement as a cancer suppressor, while others as cancer promoter [15,16]. The activation of markers involved in regulated necrosis pathways, such as ROS production, inhibition of tumor immunity, and activation of inflammation, leads to cancer progression through angiogenesis and metastasis. However, specific markers of necroptosis remain to be identified. Analysis of necroptosis, especially in vivo and in human tissue samples, is prevented by the lack of molecular markers [15,17].

In this review, we focus on the correlation between the necroptosis process and its involvement in diseases, paying attention on the biological role of RIPK1. RIPK1 expression is deregulated in certain types of human cancer tissues. Due to its involvement in activating several signaling cascades, it seems to be a good candidate for future investigation [16,18].

## 2. RIPKs Family

RIP kinases are a group of seven threonine/serine protein kinases, which share a homologous KD. RIPKs are involved in immune response, inflammation and cell death processes as sensor of both intracellular and extracellular stimuli but each member of the RIPK family participates in a specific process according to their functional domains [19]. Structurally, RIPK1, -2, -4, and -5 have an intermediate domain but only RIPK1 has a RHIM in the intermediate domain capable of interacting with the same RHIM motif of RIPK3 present at the C-terminus [20]. RIPK2 has a CARD, while RIPK4 and RIPK5 are characterized by ANK domain ankyrin repeats in their C-terminus [20]. The ANK domain is also observed in RIPK6 but it shares with RIPK7 the presence of LRR motifs and Roc/COR domains. RIPK7 harbors a WD40 domain at the C-terminal domain as well (Figure 2) [20]. While the RIPK1-3 proteins are well described, there are currently few studies on RIPK4-6. Concerning RIPK7’s biological function, it has been identified in Parkinson’s disease and as a mediator of immunity to pathogens [20,21].

Within the RIPK family, RIPK1 was the protein studied first. It has a length of 671 amino acids and was discovered following experiments based on protein-protein interactions in yeast [22]. Using genetic selection, RIPK1 was shown to bind to the intracellular domain of Fas/APO-1 (CD95) and weakly to TNFR1. These findings stimulated further research on its role in response to cell death [22]. In contrast to another RIPK family member, RIPK3, and mixed lineage kinase domain-like (MLKL) that shuttle between the cytoplasm and the nucleus, RIPK1 is constitutively present in both compartments [23].

RIPK1 shows a strong homology with the serine/threonine and tyrosine kinases and it shares a high percentage of C-terminal sequence homology with mouse [22]. RIPK1 is involved in the apoptotic process and it plays an important role in response to cellular stress and is therefore a key regulator of several signaling pathways [19]. More recently, the kinase activity of RIPK1 has been associated to increased production of IL-1α levels and to autocrine production of TNF from TNF-treated cells emphasizing its involvement in inflammation [24].

RIPK1 associates with TRIF via the RHIM domain upon TLR3 and TLR4 activation [10,25,26]. With its DD at the C-terminal, RIPK1 can be activated by several death receptors such as TNFR1, Fas, TRAILR1, and TRAILR2 [20]. After TNF-α stimulation RIPK1 can activate multiple signaling pathways resulting in cell survival or cell death [19]. Each pathway is characterized by the formation of a specific molecular complex.

During activation of TNFR1-signalling pathways by trimerized TNF-α, in the pro-survival complex, named Complex I, RIPK1 is associated with TRADD, TRAF1/2/5, cIAP1/2, LUBAC, NEMO, and deubiquitinating enzymes. RIPK1 is rapidly polyubiquitinated on K63-linked and M1-linked ubiquitin chains mediated by cIAP1/2 and LUBAC, which in turn recruit TAB2/3 and NEMO respectively. NEMO recruitment in the complex I is mostly, but not exclusively, M1-ubiquitin dependent, and mediates TBK1 and IKKε activation whose recruitment is also allowed by LUBAC activity [27]. TBK1 and IKKε phosphorylate RIPK1 on multiple residues preventing RIPK1-dependent cell death [27,28]. Additionally, K63 ubiquitination on RIPK1 mediates TAB2/3 recruitment facilitating TAK1 activation, that phosphorylates IKKα/β and RIPK1 [27]. The formation of the IKKs complex promote the activation of NF-ĸB and MAPK signaling pathways suppressing RIPK1-mediated necroptosis and apoptosis [19,20]. Following cIAPs degradation by small pharmacological compounds, deubiquitinase enzymes negatively affect the ubiquitination of RIPK1 by promoting the formation of a new molecular complex, complex II, also referred as ripoptosome [29]. In ripoptosome RIPK1 associates with TRADD, FADD, and pro-caspase 8 forming Complex IIa where activated caspase-8 cleaves RIPK1 and RIPK3 to promote RIA [30]. Alternatively, RIPK1 can form complex IIb. In conditions where complex I is inhibited or absent, activated RIPK1 can interact with FADD and caspase-8 to mediate RDA [31]. If caspase-8 is inhibited or absent and RIPK3 and MLKL expression levels are high in the cells, RIPK1 can start a series of auto and trans-phosphorylation events with RIPK3 which lead to MLKL activation to form the downstream necrosome complex beginning the necroptosis process [32].

RIPK2, through the CARD domain, interacts with NOD1 and NOD2 upon stimulation with specific bacterial fragments, which leads to NF-ĸB gene or MAPK pathway activation. Moreover, a truncated form of RIPK2 has been identified designated RIPK2-β which has only a portion of the N-terminal kinase domain and has lost the intermediate region and C-terminal CARD domain thus impairing its kinase activity [33].

Like RIPK1, RIPK3 is involved in cellular programmed necrosis. RIPK3 drives the cell death process by interacting via the RHIM motif to form heterodimeric structures with classical characteristics of β-amyloids which are crucial to mediate TNF-induced necroptosis [34]. RIPK3 controls necroptosis upon activation of cell death receptors and virus stimulation but it can also regulate inflammasome and, in addition, it directly interacts with different enzymes of the metabolic pathway after pronecrotic stimuli [20]. RIPK1 positively and negatively controls RIPK3 activation and the necroptosis processes [35]. In particular, RIPK1 is able to prevent the spontaneous activation of RIPK3 in the cytosol by negative regulation through the recruitment of caspase-8/FLIP complexes and cIAP-mediated degradation [36].

RIPK4 is involved in NF-ĸB and JNK activation since the kinase physically and functionally binds different TRAF-dependent pathways leading to the activation of NF-ĸB. Moreover, upon pro-apoptotic stimuli RIPK4 is cleaved by caspase-8 at Asp340 and Asp378 in the intermediate domain leading to blockage of NF-ĸB anti-apoptotic signals [37]. RIPK5 can induce both caspase-dependent and caspase-independent cell death pathways but much remains to be assessed regarding its physio-pathological role [38]. RIPK6 and RIPK7 show different structures than the other members. RIPK6 contains ankyrin repeat domains, which are not observed in RIPK7, but they have other domains in common such as leucine-rich repeat and Ros of complex proteins/C-terminal of Roc domains. RIPK7 further has WD40 repeats in its C terminus [20]. Based on structural studies, given their high similarity they could have evolutionarily conserved functions demonstrated by the presence of orthologous RIPK6 and RIPK7 in *Drosophila* and *Caenorhabditis elegans* [20]. However, their biological role needs to be further investigated. Here, it is interesting to note that several studies focus on pathogenetic variants of RIPK7 that are associated with the development of Parkinson’s disease [20,39].

## 3. Biological Relevance of RIPK1

RIPK1 is involved in kinase dependent and independent functions to regulate homeostasis [40]. The role of RIPK1 is widely discussed in inflammatory pathologies and diseases but little is known about its biological relevance in cancer [16]. Experimental mutations associated with RIPK1 in the different domains affect homo/hetero-dimerization, phosphorylation, and ubiquitination and can positively and/or negatively modulate cell death and cell survival [41,42]. Here, RIPK1 kinase activity that is not essential for pro-survival signaling is required to activate cell death and inflammation [40,43].

Mouse studies have demonstrated that RIPK1 is essential to regulate perinatal survival while mice *Ripk1^−/−^* die few days after birth showing severe gut disease, skin inflammation, and loss of body weight [43].

*RIPK1* is constitutively expressed in many tissues with a different degree of expression [44]. Interestingly, several mutations can lead to aberrant activation of RIPK1 [40]. Accumulating evidence suggest that mutations in *RIPK1* gene have been associated with immune and inflammatory diseases [40] (Table 1). The identified mutations underscore the importance of its role in driving the pathogenesis of various inflammatory diseases [40].

Until now, rare LoF and GoF *RIPK1* mutations have been studied in humans, mostly in the context of impaired immune system and inflammatory pathway [40]. Homozygous mutations with loss of function in RIPK1 protein were identified in four patients [45]. The mutations into the N-terminal kinase domain produce a premature stop codon and the affected patients do not express RIPK1 protein. The complete RIPK1 deficiency in humans compromises the immune system leading to an alteration in the activation of MAPK and the secretion of cytokines with predisposition to necrosis. Clinical features include recurrent infections, IBD, arthritis, and lymphopenia syndrome [45]. Hematopoietic stem cell transplantation resolved some symptoms associated to RIPK1 deficiency, representing an effective treatment for patients [45,46]. Recently, biallelic mutations with loss of function of RIPK1 protein have been identified [46]. The deregulation compromises innate and adaptive immunity with predisposition to IBD. Mostly, these are missense mutations analyzed with gene sequencing that involve amino acid mutations in the death domain of the protein which are conserved between species, while only one case presents a truncated protein [46]. Episodes of fever and lymphadenopathy were found in seven patients with heterozygous mutations in the highly conserved RIPK1 domain D324N, D324H, D324Y, and D324V. This residue is essential for caspase-8 and caspase-6 cleavage, so the gain of function mutations can affect inflammation with inappropriate activation of the inflammatory pathway [47,56]. Another genomic analysis reported a new mutation in the RIPK1 protein found in a Chinese boy with immune system deregulation and intestinal inflammation already from the first months of life. The sequenced genome identified two heterozygous mutations. While one of these mutations is a missense, the other one could be deleterious because it resulted in an early stop codon of the protein in position 333 leading to nonsense-mediated decay or creating a premature truncated protein [57].

### Mutagenesis Studies in RIPK1 Domains

To evaluate the biological relevance of RIPK1 and support the idea of RIPK1 as a therapeutic target in TNF-signaling pathway, several mutagenesis studies in RIPK1 domains have been conducted (Figure 3).

While *Ripk1^−/−^* mice shortly die after birth, mice with a RIPK1 inactive catalytic domain harboring the D138N mutation are protected from the shock induced by TNF treatment showing less hypothermia and mortality [36,50]. These results, already observed in MEFs and macrophages, indicate that the kinase activity is not essential for mice survival but it is crucial for TNF-induced necroptosis [50]. The site K45 in RIPK1 kinase domain has a key role in promoting necroptosis. Indeed, it has been clearly demonstrated that K45A mutation determines attenuated necroptosis upon stimulation by TNF-α, LPS, and IFN-β in the absence of caspase signaling. Additionally, RIPK1 K45A mice exhibited a reduced inflammatory response following endotoxic shock [48]. During the catalytic reaction, RIPK1-KD is involved in post-translational modifications such as phosphorylation of specific residues regulating its cytotoxic potential [58]. Within the KD, the conserved residue S161 allows RIPK1 to bind FADD and RIPK3 to induce cell death after TNF stimulation [59]. RIPK1 autophosphorylation on S161 is related to pro-necroptotic function and the substitution with alanine downregulates the RIPK1 kinase activity of about 20% [60]. However, mutagenesis studies revealed that other sites could positively or negatively regulate necroptosis. Serine/threonine or tyrosine residues are important sites in the KD of RIPK1 since these are acceptors of phosphate groups, hence amino acid substitutions can affect RIPK1 functions [58]. An important conserved residue in the catalytic domain of RIPK1 is serine 25 (S25), which is phosphorylated by the IKKα/β in TNFR1 complex I, repressing the kinase activity of RIPK1 with consequent inhibition of apoptosis/necroptosis [49]. Interestingly, phospho-mimetic S25D mutation in MEFs and in mice revealed protection against TNF stimulation in condition of IKK inhibition highlighting its protective role [49]. Serine 89 also is a novel regulatory residue on RIPK1 which might control programmed necrosis [60]. Reconstituted Jurkat cells expressing S89A-RIPK1 showed increased levels of TNF-induced programmed necrosis compared to wild type RIPK1 or S89D-RIPK1 revealing that phosphorylation on S89 limits RIPK1 activity [60]. Notably, several other residues have been reported to be phosphorylated by some kinases preventing RIPK1 autophosphorylation and the consequent formation of complex II [61,62,63]. TBK1 and IKKε prevent TNF-induced cell death blocking RIPK1 autoactivation and consequent complex-II formation [27,28]. It has been demonstrated that TBK1 blocks trans activation of RIPK1 through the phosphorylation on amino acid residues T189 in human and T190 in murine [28]. Additionally, IKKα, IKKβ, and MK2—downstream players of TAK1—mediate the phosphorylation that regulate TNF-mediated RIPK1-dependent cell death [27]. Upon TNF treatment, MAPK14 (p. 38) and its substrate MK2 play an essential role in mouse and human RIPK1 activation. Within the intermediate domain S321 and S336 in MEFs are phosphorylated residues by MK2 leading to a low probability to form complex II without preventing TNF-induced activation of NF-ĸB [62,63]. The importance of the molecular function of the ID was also demonstrated by the expression of a mutant form of RIPK1 lacking the intermediate domain (RIPK1ΔID) in L929sA cells. The loss of ID domain in RIPK1 induces a switch from TNF-induced necroptosis to RDA because it breaks down the interaction with RIPK3 to form the necrosome [64]. Moreover, RIPK1ΔID expression excludes TRADD to form the complex II, suggesting that RDA is different from classic apoptosis (RIA) in which TRADD needs to recruit FADD and Caspase-8 for death induction [64]. In vitro mutagenesis studies have identified how lysine residue K377 in the intermediate domain is a functional site for the ubiquitination regulating the pathway between cell death and survival after TNF-α stimulation. Mutation in this residue, K377R, prevents ubiquitination and therefore the interaction between RIPK1 and TAK1 with a consequent reduction in the percentage of survival, but does not influence the association with the NEMO subunit, highlighting the role of RIPK1 ubiquitination for NF-ĸB activation [65]. K377 in human RIPK1 corresponds to K376 in murine RIPK1 and is the residue for K63-linked ubiquitination, fundamental for activation of the cell death program [51]. The generation of RIPK1 knock-in mice in which the lysine residue K376 is replaced by arginine (R) lead to an early embryonic lethality compared to *Ripk1^−/−^* mice with perinatal death [51]. To further evaluate which molecular mechanisms were behind this effect, MEFs were generated with the same alteration. Following TNF-α stimulation, MEFs with homozygous K376R mutation showed high level of RIPK1-dependent apoptosis and necroptosis due to the increase of RIPK1 kinase activity likely caused by the reduced K63-linked ubiquitination with consequent inhibition of the Complex I formation [51]. In the intermediate domain, a potential caspase cleavage site is present at D324 that may modulate RIPK1 activation during embryonic development [52]. Caspase-8 activation through the adaptor FADD may lead to impair the RIPK1 D324 cleavage site protecting embryonic cells by apoptosis and necroptosis. Interestingly, replacing D324 with alanine (A) determines an early embryonic lethality in mice due to RIPK3-dependent necroptosis [52]. In the context of the intermediate domain, D325 in RIPK1 is the residue cleaved by caspase-8 to prevent cell death in development [53]. MEF cells expressing RIPK1 mutation D325A showed increased TNF-induced cell death compared to wild type. These data have been confirmed in mice in which the mutated residue in RIPK1 cannot be cleaved, generating death during embryogenesis [53]. Upon RIPK1 activation the negative charges generated by phosphorylation of multiple serine and threonine residues of the kinase domain permit RHIM-RHIM motif association between RIPK1 and RIPK3 to form an amyloid structure required for necrosome activation [60]. In particular, the interaction requires the IQIG consensus sequences of RIPK1 and VQVG of RIPK3 [66]. Mutated forms of the RHIM motif can impair necroptosis. For example, spectroscopy analysis reveal that the A543D mutation prevents the formation of the RIPK1-RIPK3 hetero-amyloid and initiation of necroptosis [66]. Changes in RIPK1 RHIM motif residues IQIG mutated to AAAA in mice leads to an early death and histologic lesions that can be prevented by RIPK3, MLKL, or ZBP1 deficiency [54]. The death domain of RIPK1 has a functional role promoting RIPK1 activation under necroptosis and RDA conditions both in vitro and in vivo. Indeed, the mutation of the conserved residue of lysine to arginine, K599R in human and K584R in murine, affects RIPK1 homodimerization leading to block RIPK1 activation in the cytosol and the formation of complex II [55]. In addition, structural studies revealed that mutations M637A and R638A in RIPK1 DD disrupt the stable interactions among proteins containing the DD such as TNFR1 and TRADD [67]. Together, these mutagenesis studies allowed investigation and elucidation of the functional role of RIPK1 in activating the different cellular pathways through the recruitment of several molecular players. These strategies enabled to study RIPK1 behavior in the contexts of cell survival and cell death, helping the study and the treatment of several diseases including cancers.

## 4. Regulated Necrosis: Necroptosis

Recently, different forms of regulated necrosis were studied, expanding the network of non-apoptotic cell death [11]. Necroptosis was the first programmed necrosis discovered, prominently found when the apoptotic pathway is altered or inhibited. As a mechanism of cell death, necroptosis participates in development programs to ensure the maintenance of cellular homeostasis [11]. Ample evidence supports the idea that it evolved in order to provide a protection against infections [3,13]. Morphological changes observed in necroptotic cells are cytoplasm and organelle swelling, vacuolization, membrane rupture and release of cytoplasmic content in the extracellular microenvironment [15]. Moreover, at biochemical level mitochondrial fission, ROS production, ATP depletion, and calcium and sodium intake have been observed [17]. Molecularly, it is characterized by caspase inactivation and controlled by the kinase activity of members of receptor interacting serine/threonine kinase family. Indeed, the core of the necroptosis machinery is constituted by RIPK1, RIPK3, and MLKL [25].

Necroptosis can be activated by several stimuli, in particular by the pro-inflammatory cytokine TNF-α [25]. Here, when TNF-α binds its receptor different cell-response are activated such as: (i) cell survival, with inflammatory cytokines production following NF-ĸB activation; (ii) RIA, mediated by FADD and Caspase-8 interaction after NF-ĸB pathway inhibition; (iii) RDA, after Smac mimetic molecule activity, which promotes a reduction in RIPK1 ubiquitination and consequent activation of Caspase-8; and (iv) cell death under apoptosis-deficient conditions via RIPK1/RIPK3/MLKL complex (which is called the necrosome) formation [30].

Other ligands may activate necroptosis as well, such as Toll-like receptors, Interferon receptors, and virus sensors [2].

Necroptosis and apoptosis are closely interconnected and the switch from one cell death modality to the other one could be mediated by TAK1 and its interactors [13,68]. TAK1 mediates RIPK3 phosphorylation leading both to necrosis without caspase activation and pro-cell survival maintenance. Upon TNF treatment, TAK1 is quickly transiently activated and deactivated by type 2 protein phosphatases such as PP2A and PP6 activity. This event is sufficient to inhibit caspase-8 and to compromise the apoptotic pathway inducing cell survival through NF-ĸB activation and antioxidant enzymes production [68]. During TAK1 deactivation, TAB2 has a functional role by recruiting PP6 in the formed complex [68]. Indeed, in TAK1-deficient cells the activation of caspase-8-mediated apoptotic pathway has been observed, while TAK1 hyperactivation induced the RIPK1-RIPK3 necrotic pathway. Furthermore, RIPK3 activation leads to TAK1 hyperactivation suggesting a positive feedforward mechanism. These results demonstrate that TAK1-RIPK1-RIPK3 cooperate in the activation of apoptotic and necroptotic pathways, but more studies are required to better define the molecular mechanisms [68].

Impairment of the NF-ĸB pathway resulting in transient RIPK1 phosphorylation in serine 321 (S321) TAK1-mediated induces the recruitment of FADD and Caspase-8 to arbitrate RIA. To emphasize the biological function of this amino acid residue, it has been shown that phosphorylation blockade occurs when serine is replaced by alanine (S321A) and consequently the RDA cascade is activated [63,69].

Some epigenetic modifications such as ubiquitination and phosphorylation have a primary role for the activation of the necrosome complex and cell death induction [70]. RIPK1 self-phosphorylation at serine 166 (S166) determines the activation of RIPK3, its oligomerization and phosphorylation that in turn can activate MLKL [4]. This MLKL oligomerized structure moves to the plasma membrane and induces its rupture with the consequent release of the intracellular content including pro-inflammatory cytokines and DAMPs into the microenvironment [41]. Moreover, phosphorylation of RIPK3 and MLKL is required for their translocation from nucleus to cytosol and for the activation of the necroptotic death pathway [71]. A key molecular event is RIPK1 ubiquitination in K63, an essential phenomenon both for survival and death signaling pathways. It is known that, following deubiquitination, RIPK1 leaves TNFR1-complex to form the necrosome ones. A recent study has shown that in this complex, RIPK1, undergoes a second step of ubiquitination performed by HOIP, a component of the LUBAC complex [72]. Additionally, following TNF-mediated necroptosis, RIPK1 was reported to be also ubiquitinated in the nucleus, suggesting RIPK3 and MLKL localization in the same cellular compartment. Consequently, the export of phospho-RIPK3 and phospho-MLKL from the nucleus to the cytosol is important for necroptotic cell death [71].

Increasing evidence also suggests a role for acetylation in the regulation of programmed necrosis in which SIRT1/2 play a crucial role in modulating RIPK1 deacetylation [73,74]. Following necrotic stimulation, RIPK1 assumes an active conformation to allow deacetylation of lysine 530 (adjacent site to RHIM) carried out by SIRT2 which is located in the pre-formed complex with RIPK3 [73]. Interestingly, the more recently identified complex RIPK1-HAT1-SIRT1/2 strengthens the important biological function of de-acetylation in modulating cell death pathways. The discovery of several acetylation sites on RIPK1 (K115 within the kinase domain; K625, K627, K642, and K648 within the RIPK1 death domain) suggests that acetylation might regulate PCD [74]. Treatment with a novel pan-SIRT inhibitor identified an increased acetylation in RIPK1 death domain on additional lysine residues (596 and 599 within the death domain) resulting in cell death modulation, strengthening how RIPK1 acetylation can control the cell death pathways in a definite manner [74].

Despite the importance of RIPK1, different studies underlined that RIPK3 and MLKL activate necroptosis in a RIPK1 independent manner [15,17,41]. Indeed, in HT-22 and L929 RIPK1 knockdown cell lines, after stimulation with TNF-α, the molecular complex triggering necroptosis is represented by RIPK3, MLKL, and TRADD. Although RIPK1 is present in small amount, it competes with the adaptor protein TRADD to bind TNFR and to activate RIPK3. However, RIPK1 knockdown cells facilitate the interaction between TRADD and RIPK3 resulting in RIPK1-independent necroptosis. In addition, TRADD promotes RIPK3 oligomerization via intramolecular reaction [75]. These events trigger MLKL activation with an increase of ROS production [75]. Other reports describe that necroptosis induction by dsRNA or LPS requires TLR-3 and -4 respectively to activate RIPK3 directly through TRIF via RHIM–RHIM interaction [16,76]. An important step mediating necroptosis under LPS stimulation is RIPK3 O-GlcNAcylation carried on by OGT [77]. The modification on the T467 residue could suppress RHIM-mediated effects, such as RIPK3–RIPK1-hetero- and RIPK3–RIPK3 homo-interaction, as well as the RIPK3 kinase activity leading to a negative regulation of necroptosis response [77]. In addition, as a result of viral infection, there is activation of RIPK3 with a DNA-dependent activator of interferon-regulatory factors through RHIM–RHIM interaction. The necroptotic sensitivity, in a RIPK1 deficiency cellular system, could be explained by IAP protein degradation which may sensitize the cells to necrosome or complex IIb (RIPK3/MLKL) formation underlying their function as necroptotic biomarkers [78].

Thus far, few reports have described the involvement of mitochondria in the necroptosis mechanism. In some cell lines, following treatment with TNF-α and caspase inhibition, the activated form of RIPK3 engages dehydrogenated pyruvate complex with a subsequent increase in aerobic respiration and generation of ROS, which in turn enhance necrosome assembly determining necroptosis activation [79]. In addition, recent data support the idea that necroptosis is not compromised after TNF-α plus zVAD-fmk stimulation in cells depleted of mitochondria defining ROS production as the consequence, but not the cause of, necroptosis [29,80].

## 5. The Dual Face of Necroptosis in Cancer

Resistance to apoptosis, due to alteration or inactivation of caspases activity is considered a hallmark of cancer [17]. Based on this evidence, necroptosis induction could be used as a strategy for cancer treatment to overcome apoptosis-resistant cancer cells [15,18]. However, the role of necroptosis in cancer is widely discussed because it can take the dual function as tumor suppressor or tumor promoter [25,30]. Necroptosis in cancer can be described as a ‘double-edged sword’ because its induction supports the death of abnormal cells leading to a good prognosis; on the other hand, in the same pathological conditions, necroptosis could determine the activation of alternative pathways and evoke inflammation, complicating cell fate and the outcome of the pathologies leading to inflammatory diseases, neurodegenerative diseases or cancer metastasis [16,17].

Various mechanisms have been proposed to elucidate the dual function. Considering the different microenvironment of each type of cancer and the pleiotropic role of its mediators, it is not possible to define universally the role of necroptosis in tumorigenesis [25]. However, hypoxia, a hallmark of solid tumor, is a valid strategy adopted by cancer cells to survive and overcome necroptosis. In this way, cancer cells can reprogram their metabolic system by decreasing the sensitivity to necroptosis [18].

An interesting role in the activation of the necroptotic cascade in tumorigenesis is attributed to the DAMPs involved, at least partially, in the inflammatory response [30,81]. The release into the microenvironment of DAMP such as HMGB1, cytokines (i.e., IL-1), ATP, RNI, ROS, and mitochondrial DNA, after necroptotic stimuli, induces inflammation, responsible for malignant transformation and tumor metastases [15,16]. However, necroptotic cells through the release of DAMP can also activate the immune system and promote tumor suppression [3,15]. Necroptotic cells present antigens to DCs which in turn activate CD8+ T cells through a phenomenon known as antigenic cross-priming [16,81]. It has been shown that the activation of T lymphocytes to organize adaptive immunity not only depends on the release of DAMPs but requires RIPK1-mediated signaling and NF-ĸB-activated transcription [82]. DAMPs release during necroptosis may be a crucial event to understanding the apparently opposite functions of the immune system in cancer immune surveillance and tumor promotion [16]. Recently, it has been hypothesized that RIPK1 may be a new immunomodulatory target for the progress of new anticancer drugs. Indeed, inhibition of RIPK1 kinase activity has been displayed to enhance antitumor immunity through modulation of tumor-associated macrophages. Hence, targeting RIPK1 kinase also contributes to the antitumor effect by contrasting the immunosuppressive necroptotic tumor microenvironment. In a pancreatic cancer model, administration of RIPK1 inhibitors has been shown to reprogram tumor-associated macrophages from a tolerogenic to an immunogenic state [83].

In the anti-tumor immune response, RIPK3 also has a pivotal role. Although previous studies have shown that RIPK3 is not involved in the activation of B cells, T cells or macrophages, recently it has been reported that RIPK3 regulates the activation of NKT cells activating the immune response and the lysis of cancer cells [84].

The dual role of necroptosis has also been associated to the metastasis process. Metastasis is the ability of cancer cells to reach a new location in the body and is the primary cause of cancer patient mortality. Although there are still only few studies, it was demonstrated that all adverse phenomena, such as activation of the immune system, hypoxia, DNA mutations and cellular overproduction of ROS, can trigger necroptosis of metastatic cells [7]. On the other hands, conflicting evidences suggest necroptosis as a promoter of metastasis. Indeed, it has been reported that the death receptor 6 on the surface of endothelial cells binds its ligand amyloid precursor protein to promote endothelial cell death and tumor cell extravasation [30]. Moreover, RIPK1/RIPK3 promote vascular permeability to mediate tumor cells extravasation independent of necroptosis since they allow heat shock protein 27 phosphorylation in lung endothelial cells upon permeability factor treatment (VEGF-A, VEGF-B, FGF-b) [85].

This evidence reveals a controversial role of necroptosis in tumorigenesis. By exploiting the already known therapeutic strategies such as immunotherapy, radiotherapy, or the administration of chemotherapeutic agents, it will be interesting to set new approaches to possibly targeting necroptosis for cancer treatment. In this context, RIPK1 activity modulation may be a promising therapeutic approach for novel therapeutic strategies. RIPK1 inhibitors could be useful alternative treatments for patients who are non-responders to anti-TNF treatment. Several multitargeting tyrosine kinase inhibitors with a broad spectrum have been approved by FDA both for solid and hematological cancers. Nowadays, only few small molecules have been characterized to specifically target RIPK1, and they are in clinical trials phase I/II for the treatment of inflammatory and degenerative diseases. Clinical trials of RIPK1 inhibitors for tumors therapy have not been successful. To date, only one compound that targets RIPK1 has been characterized for treatment of pancreatic cancer and it is in phases I/II [83]. Therefore, further efforts understanding the precise role of RIPK1 activity in cancer models will be critical for the development of new therapies [83].

## 6. Involvement of Necroptotic Mediators in Cancer

It is not surprising that RIPK1, RIPK3, and MLKL in cancer play a central role in modulating necroptosis. Functional mutations in the necroptotic machinery can compromise the death of cancer cells and influence the prognosis due to changes in interactions between RIPKs and other proteins [18]. Upregulation and downregulation of key necroptotic members have been found in many cancer types [16] (Table 2).

Over the years, interest in the correlation between gene mutations associated with RIPK1 and cancer has grown. Furthermore as reported by the COSMIC database about the somatic mutations in cancer, *RIPK1* gene is not significantly mutated in a specific tissue, rather, the identified mutations are: about 36% missense substitutions, 10% synonymous substitutions, 5% nonsense substitutions, 0.4% frameshift insertions, 0.2% frameshift deletions, and 9% of mutations without detailed information available [99]. *Ripk1* belongs to those genes with a part probably implicated in cancer and the ICGC database reported genomic and proteomic mutations in cancer distribution (https://dcc.icgc.org/). In general, RIPK1 maintains normal expression levels in many cancer types but in glioblastoma or lung cancer its expression has been found to be upregulated, affecting cancer prognosis negatively [16]. RIPK1 protein expression seems to be higher in glioblastoma grade IV, the most common adult malignant brain tumor, compared to lower grade glioma (I–III). RIPK1 overexpression with a consequent activation of NF-ĸB, upregulation of MDM2 and inhibition of p53 pathway lead to a worse prognosis, resistance to DNA damage and malignant phenotype [91]. According to the tumor-promoting role of RIPK1 in glioblastoma, a recent study suggests the oncogenic function of RIPK1 in the lung epithelial cells that have acquired genetic mutations and epigenetic modifications caused by carcinogens [94]. An increase in RIPK1 expression induced by cigarette smoke carcinogens suppresses ROS accumulation and MAPK-mediated cytotoxicity in DNA-damaged bronchial epithelial cells by facilitating malignant transformation [94]. RIPK1 is also a critical regulator of hepatocyte survival that cooperates with NF-ĸB to control TNFR1-dependent and -independent chronic liver inflammation and cancer [93,100]. Under physiological conditions, RIPK1 synergizes with NF-ĸB to prevent hepatocyte apoptosis, chronic liver disease and cancer. However, in the absence of NEMO, RIPK1 activity drives hepatocyte apoptosis in a kinase-dependent manner, resulting in HCC development [100]. Additionally, low expression of RIPK1 and TRAF2 in HCC patients undergoing liver resection or transplantation predicted poor prognosis [93].

The tumor stage can also influence RIPK1 expression levels, for example in HNSCC the metastatic stage shows reduced transcriptional and protein expression levels of RIPK1 compared to primary tumor and this could be associated to epigenetic events such as the methylation status in the RIPK1 promoter. Hypermethylation of a specific site reduces the binding of the transcription factor ARID3A to the promoter of *Ripk1* resulting in a downregulation of the protein with compromised cell integrity [92].

Fusion genes have a significant evolution in the architecture of a gene and play a key role in tumorigenesis since these represent an important class of somatic alterations in cancer [101]. *RIPK1* belongs to the genes involved in translocations which are found in adenocarcinoma of breast, prostate, and ovary (Table 3). *RIPK1* has been found also in Astrocytoma, Grade III-IV/glioblastoma in which translocation t(6;6) (p25:p25) occurs. In order to suggest therapeutic approaches and predict kinases inhibitors it is interesting to unravel the molecular roles of these fusion transcripts involving protein kinase genes [101].

Finally, *RIPK1* also belongs to a set of 21 genes that are aberrant expressed and revealed to be more likely involved in the metastatic process of monosomy 3 in uveal melanomas [105].

In addition to RIPK1, the necroptotic mediators RIPK3 and MLKL are also involved in cancer and have influence on prognosis. It is already known that most cancer cells lose RIPK3 expression, so the downregulation or deletion of RIPK3 during tumorigenesis accompanied by necroptosis resistance determines a poor prognosis [16,106]. This deregulation could be mediated by oncogenes such as BRAF and AXL/TYRO3 able to control one of the transcription factors (Sp1) of RIPK3 and the promoter by methylation state [107,108]. A hypo-methylation of RIPK3 promoter that restores the normal expression of the kinase leads to an increase in sensitivity to chemotherapy by improving anticancer treatment [106]. Interestingly, in some colon cancer cell lines the mRNA expression levels of both *RIPK1* and *RIPK3* decrease due to hypoxia but not by promoter methylation status leading to a worse prognosis [109]. Additionally, RIPK3 is downregulated in breast, colorectal, melanoma, and AML cancers [16]. Although the mechanisms are still unclear, RIPK3 and MLKL deficient breast cancer cells are related to a reduced expression of the genes involved in interferon-α and interferon-γ responses [87]. RNA-Seq analysis using CD34^+^ bone marrow cells from patients with myelodysplastic syndromes or chronic myelomonocytic leukemia revealed overexpression of MLKL and its association with the severity of anemia. The increased level of RIPK1 expression supports its role as a mediator of inflammation, defining it as a predictor of worse overall survival although the mechanism remains unclear [110]. AML patients with CD34^+^ leukemia cells show reduced RIPK3 expression while the expression of RIPK1 is not affected leading to suppressed apoptosis, necroptosis, and NF-ĸB pathway [86]. Moreover, an additional study suggested that RIPK1/RIPK3 inhibition may be an effective treatment for AML patients when combined with specific chimeric antigen receptor T cells (that express high levels of IFN-γ) or other differentiation inducers in order to repress leukemogenic capacity of AML cells [111]. Tumor progression and reduced survival in patients with early-stage pancreatic adenocarcinoma, gastric, ovarian, and cervical squamous cancer is associated to low MLKL expression [89,90]. The downregulated MLKL expression could lead to inhibition of the necroptotic phenomenon and therefore determine a poor prognosis underlying its role as prognostic biomarker in cancer. In contrast RIPK1, RIPK3, and MLKL are widely expressed in human pancreatic ductal adenocarcinoma associated with robust manifestation of chemokines resulting in tumor promotion [98].

## 7. Conclusions

Cell death has a great impact in the regulation of physiological and pathological phenomena. The recent discovery of a new form of programmed necrosis has opened the way to develop pro-necroptotic drugs for anti-cancer therapy. More studies are needed to contextualize the role of necroptosis in different types of cancer. Since necroptotic phenomenon can take the dual function of tumor suppressor and tumor promoter, it could be interesting to manipulate its mediators to direct cancer cells towards death by improving patient survival. Using the members of the necroptotic system as biomarkers in cancer can be useful for prognostic information. Many in vitro studies have been conducted so far, so more efforts are needed to explore necroptotic mechanisms in vivo to achieve more efficient anticancer therapies.

## Figures and Tables

**Figure 1 cancers-13-00684-f001:**
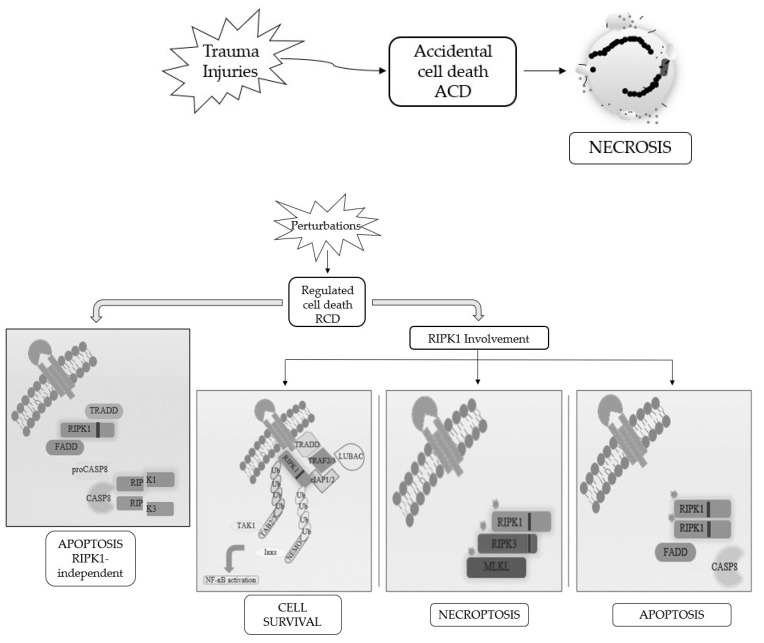
Schematic representation of the best-known cell death modalities with different molecular features.

**Figure 2 cancers-13-00684-f002:**
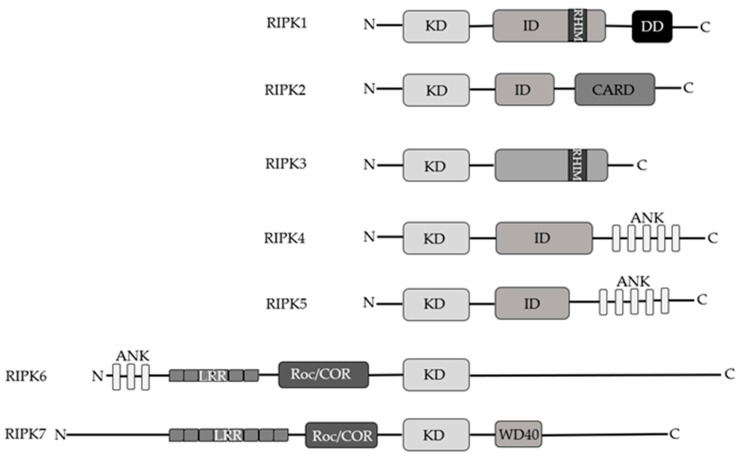
Schematic representation of RIP kinases family’s members.

**Figure 3 cancers-13-00684-f003:**
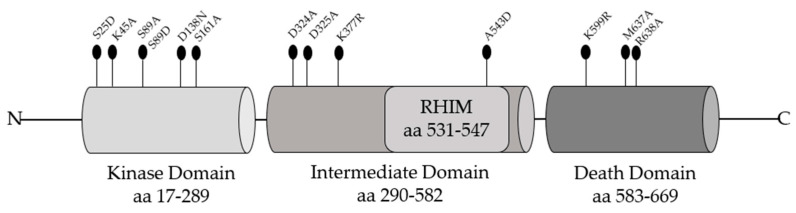
Aminoacidic substitutions in RIPK1 domains useful to elucidate its functional roles.

**Table 1 cancers-13-00684-t001:** RIPK1 in vivo mutations and its main effects.

Species	Mutations	RIPK1 Site	Effects	References
Human	Loss of Function	Protein impairment	Combined Immunodeficiency and IBD	[45,46]
Human	D324N D324H D324Y D324V	Intermediate domain	Autoinflammation with Episodic Fever and Lymphadenopathy	[46,47]
Mouse	Ripk1-deficiency		Perinatal death	[43]
Mouse	K45A	Kinase domain	Necroptosis resistance	[48]
Mouse	S25D	Kinase domain	Protection from TNF-induced lethal shock	[49]
Mouse	D138N	Kinase domain	Protection against TNF shock	[50]
Mouse	K376R/K376R K376R/-	Intermediate domain	Embryonic death Severe inflammation	[51]
Mouse	D324A	Intermediate domain	Midgestational death	[52]
Mouse	D325A D325A/+	Intermediate domain	Gestational death Vitality with increased susceptibility TNF-death	[53]
Mouse	IQIG→AAAA	RHIM motif	Perinatal lethality	[54]
Mouse	K584R	Death domain	Protection against TNFα-induced SIRS	[55]

**Table 2 cancers-13-00684-t002:** Deregulated expression of necroptotic factors.

Cancer Model	Dysregulated Expression	References
AML	RIPK3 downregulation	[86]
Breast	RIPK3 and MLKL downregulation	[87]
Colorectal	RIPK3 downregulation	[88]
Cervical Squamous Cell Carcinoma	MLKL downregulation	[89]
Gastric	MLKL downregulation	[90]
Glioblastoma	RIPK1 upregulation	[91]
Head and Neck Squamous Cell Carcinoma	RIPK1 downregulation	[92]
Liver	RIPK1 downregulation	[93]
Lung	RIPK1 upregulation	[94]
Ovarian	MLKL downregulation	[95]
Melanoma	RIPK3 downregulation	[96]
Pancreatic adenocarcinoma early-stage	MLKL downregulation	[97]
Pancreatic ductal adenocarcinoma	RIPK1, RIPK3, and MLKL upregulation	[98]

**Table 3 cancers-13-00684-t003:** RIPK1 fusion gene involved in tumorigenesis

Morphology-Topography	Genes	Abnormality	References
Breast Adenocarcinoma	RIPK1/BCKDHB	t(6;6) (p25;q14)	[101]
Breast Adenocarcinoma	RIPK1/NQO2	t(6;6) (p25;p25)	[102]
Prostate Adenocarcinoma	FARS2/RIPK1	t(6;6) (p25;p25)	[101,103]
Brain Astrocytoma, Grade III-IV/Glioblastoma	RIPK1/SERPINB9	t(6;6) (p25;p25)	[104]
Ovary Adenocarcinoma	RIPK1/TTC27	t(2;6) (p22;p25)	[101]

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
