# Peer review of "The Role of Necroptosis: Biological Relevance and Its Involvement in Cancer"

_cancers, 2021, doi:10.3390/cancers13040684_

Round 1

Reviewer 1 Report

In their manuscript entitled « The role of necroptosis: biological relevance and its involvement in cancer”, the authors first introduce the different types of cell death. Then, the authors describe the structure and main functions of the members of the RIPK family before focusing on the biological functions of RIPK1, the role of necroptosis and of necroptosis mediators in cancer. The review is overall nicely written and comprehensive. I only have minor comments (see below) aimed to help the authors further improve this review. A couple of typos should also be corrected.

Lines 73-76, the authors state that “that specific markers of necroptosis such as reactive oxygen species (ROS) production, inhibition of tumor immunity and activation of inflammation contribute to cancer progression [16,17]. ». These are not specific markers of necroptosis (for example, ferroptosis is also accompanied by ROS production, pyroptosis is also pro-inflammatory, etc..). So, this sentence should be modified or this statement more precise.

Lines 83-84: I think there is mistake on Figure 1, apoptosis and necroptosis should be inverted.

Line 103: It might be good to mention which protein-protein interaction where searched for in this article.

Line 123: The authors should mention that what is described in this paragraph applies to TNFR1 signalling.

Line 127: NEMO is also recruited by non-linear chains (but linear chains vastly increase its recruitment), so the statement used by the authors is not exact here. It might also be worth mentioning the recruitment of IKKe and TBK1 through NEMO here (papers from Lafont E et al, 2018, Nat Cell Biol and Xu D et al, 2018, Cell), are these are important regulators of RIPK1.

Line 130: TAB/TAK mediates the MAPK activation rather than NEMO. I am also not sure that the “following cIAP degradation” is correct: does this occur naturally after a while upon TNF stimulation or do the authors mean if it is induced pharmacologically?

Lines 162-170: RIP7, RIP6 ->RIPK6, RIPK7.

Lines 221-223: I am not sure what the author mean here, I think this should be rephrased.

Pages 6-7, regarding the phosphorylation of RIPK1, I think the two studies mentioned earlier with regard to TBK1/IKKe regulation of RIPK1 by phosphorylation should be included here, as well as the three studies demonstrating that MK2 can phosphorylate and thus regulate RIPK1 (Dondelinger Y et al, Nat Cell Biol, 2017; Jaco etal, Mol Cell, 2017 and Menon et al, Nat Cell Biol, 2017).

Line 320: How is TAK1 activation proposed to limit caspase-8 activation?

Line 342: I think the authros should remind the authors that linear ubiquitination in Complex I conversely prevents RIPK1 activation here.

Line 361: what is the evidence provided in this article that this is really RIPK1-independent? (if this are knock-down cells, whatever small amount of RIPK1 might still do the job…).

The recent study from Li X. et al (Immunity 2019) on modulation of RIPK3 by O-GlcNAc should be commented in this review as well. There are quite a number of studies linking RIPK1 and hepatocarcinogenesis (for example work from M. Pasparakis and T. Luedde’s labs, among others) which would fit in this review.

Author Response

Reviewer 1

Q1: Lines 73-76, the authors state that “that specific markers of necroptosis such as reactive oxygen species (ROS) production, inhibition of tumor immunity and activation of inflammation contribute to cancer progression [16,17]». These are not specific markers of necroptosis (for example, ferroptosis is also accompanied by ROS production, pyroptosis is also pro-inflammatory, etc..). So, this sentence should be modified or this statement more precise.

A1: Thanks for your observation. We have modified the sentence in the main text (page 6; lines 77-86)

Q2: Lines 83-84: I think there is mistake on Figure 1, apoptosis and necroptosis should be inverted.

A2: We apologize for the mistake; we have corrected the figure. Please see now the new version of figure 1

Q3: Line 103: It might be good to mention which protein-protein interaction where searched for in this article.

A3: Thanks for the suggestion. We have better explained the RIPK1 protein interactions. Please see page 8; lines121-124

A4: Line 123: The authors should mention that what is described in this paragraph applies to TNFR1 signalling.

Q4: Thanks for the comment. We have modified the text. (Page 8; lane139)

Q5: Line 127: NEMO is also recruited by non-linear chains (but linear chains vastly increase its recruitment), so the statement used by the authors is not exact here. It might also be worth mentioning the recruitment of IKKe and TBK1 through NEMO here (papers from Lafont E et al, 2018, Nat Cell Biol and Xu D et al, 2018, Cell), are these are important regulators of RIPK1.

A5: Thanks for the useful comment. We have better explained the role of IKKe, TBK1 and NEMO in RIPK1 complex, taking in consideration the suggested articles. (Page 8; lines 141-149)

Q6: Line 130: TAB/TAK mediates the MAPK activation rather than NEMO. I am also not sure that the “following cIAP degradation” is correct: does this occur naturally after a while upon TNF stimulation or do the authors mean if it is induced pharmacologically?

A6: Thanks for the comment. We have better explained the involvement and the role of MAPK signalling (page 8; lines 147-149). We also have better clarify that cIAP degradation is pharmacologically induced (page 8; lines 149-150)

Q7: Lines 162-170: RIP7, RIP6 ->RIPK6, RIPK7.

A7: we have corrected the text accordingly (page 9, lines:178-186).

Q8: Lines 221-223: I am not sure what the author mean here, I think this should be rephrased.

A8: thanks for the suggestion. We have rephrased the sentence. (Page11; lines 270-273).

Q9: Pages 6-7, regarding the phosphorylation of RIPK1, I think the two studies mentioned earlier with regard to TBK1/IKKe regulation of RIPK1 by phosphorylation should be included here, as well as the three studies demonstrating that MK2 can phosphorylate and thus regulate RIPK1 (Dondelinger Y et al, Nat Cell Biol, 2017; Jaco etal, Mol Cell, 2017 and Menon et al, Nat Cell Biol, 2017).

A9: Thanks for the useful comment. We have improved the main text following your suggestions, and the references have been added in the bibliography section (Pages 11-12; lines: 289-299)

Q10: Line 320: How is TAK1 activation proposed to limit caspase-8 activation?

A10: Thanks for your suggestion. We have better explained TAK1 activity. (Page 13; lines: 368-379).

Q11: Line 342: I think the authors should remind the authors that linear ubiquitination in Complex I conversely prevents RIPK1 activation here.

A11: Thanks. Following your useful suggestion, we have improved the main text. (Pages 13-14; lines: 392-396).

Q12: Line 361: what is the evidence provided in this article that this is really RIPK1-independent? (if this are knock-down cells, whatever small amount of RIPK1 might still do the job…).

A12: Thanks. We have better explained our sentence following your comment. (Page 14; lines: 413-418).

Q13: The recent study from Li X. et al (Immunity 2019) on modulation of RIPK3 by O-GlcNAc should be commented in this review as well. There are quite a number of studies linking RIPK1 and hepatocarcinogenesis (for example work from M. Pasparakis and T. Luedde’s labs, among others) which would fit in this review.

A13: Thanks for the useful comment. Following your suggested paper, we have added in the texts RIPK3 modulation by O-GlcNAc. (Page 14; lines: 412-425). Moreover, concerning the involvement of RIPK1 in hepatocarcinogenesis, we have now improved the text. Please see page 17; lines: 524-531.

Reviewer 2 Report

Extensive review on a interesting topic. Few minor points:

Ensure in final manuscript that figures/tables are on one page for ease of reading

List of abbreviations needs to be included

English grammar/syntax needs to be checked in places. For instance use of capital letters and file attached highlights particular areas on which to focus.

Mention how many amino acids are in RIPKI and it would be good if possible to include a schematic highlighting where the mutations you refer to in text are in the protein sequence

Concluding sentences needed at end of mutagenesis studies in RIPK1 section - what do these studies show overall regarding the role of RIPK1

Author Response

Reviewer 2

Q1: Ensure in final manuscript that figures/tables are on one page for ease of reading

A1: Thanks for the suggestion. We have corrected the manuscript.

Q2: List of abbreviations needs to be included

A2: Thanks for the useful comment. A list of abbreviation is now present in the manuscript.

Q3: English grammar/syntax needs to be checked in places. For instance use of capital letters and file attached highlights particular areas on which to focus.

A3: Thanks for the useful comment we have corrected the text accordingly.

Q4: Mention how many amino acids are in RIPKI and it would be good if possible to include a schematic highlighting where the mutations you refer to in text are in the protein sequence

A4: Thanks. We have included this part in the text. Please see page 8; lane 121. Moreover, we also have added a figure showing all mentioned RIPK1 mutations. (Please see Figure 3)

Q5: Concluding sentences needed at end of mutagenesis studies in RIPK1 section - what do these studies show overall regarding the role of RIPK1

A5: thanks for the suggestion. We have now added a final comment to better explain the aim of mutagenesis studies. (Pages 12-13; lines: 340-344)

Reviewer 3 Report

The review is well organized and written.  The authors state that targeting necroptosis may be a therapeutic strategy as anticancer therapies. However, there was no discussion on how, what targets and small molecule inhibitors tested pre-clinically and/or the clinic. If immune mediated killing is relevant to necroptosis then what is the role of targeting host immunity to enhance inhibiting necroptosis?

Sentence construction needs clarity.  

Author Response

Reviewer3

Q1: There was no discussion on how, what targets and small molecule inhibitors tested pre-clinically and/or the clinic.

A1: Thanks for the useful comment. We have improved the quality of our review including your suggestion. (Pages 15-16; lines: 489-498).

Q2: If immune mediated killing is relevant to necroptosis then what is the role of targeting host immunity to enhance inhibiting necroptosis?

A2: Thanks. We have added this comment in the main text. (Page 15; lines: 460-470).

Q3: Sentence construction needs clarity. 

A3: Thank you. We have revisioned the main text.  

Round 2

Reviewer 1 Report

The authors have satisfactorily adressed my previous comments.